# OpenReview forum: "LiveCodeBench Pro: How Do Olympiad Medalists Judge LLMs in Competitive Programming?"
_NeurIPS.cc/2025/Datasets_and_Benchmarks_Track — NeurIPS 2025 Datasets and Benchmarks Track poster_

### Official Review · Reviewer_NmLn · 2025-06-04

**Rating:** 4
**Confidence:** 4

**Summary:**

The authors introduce LiveCodeBench Pro, a benchmark incorporating problems from Codeforces, the ICPC series, and the IOI series. To mitigate data contamination risks, the authors evaluated state-of-the-art LLMs on problems immediately upon their release in live contests. The paper concludes with four key findings.

**Dataset Code Accessibility:**

Partly

**Dataset Code Comments:**

Please See Weaknesses.

**Ethical Considerations:**

No, there are no or only very minor ethics concerns

**Final Justification:**

The author's response has resolved many of my doubts. I feel the author's test is very useful, but the provided dataset is currently completely unusable.

**Limitations Weaknesses:**

I have several concerns regarding the dataset provided in this paper:

1. **Lack of metadata**: There appears to be no information about problem categories or tags.
2. **Limited source diversity**: The dataset only includes problems from Codeforces and QOJ.
3. **Missing ELO scores**: No difficulty rating (e.g., ELO score) is available for the problems.
4. **Absence of LLM-generated answers**: It is unclear whether large language model (LLM) outputs are included.
5. **No token-level generation data**: There is no record of token-by-token generation traces from the LLMs.
6. **Missing inference cost**: The cost (e.g., token count, latency) per LLM answer is not reported.
7. **Ambiguity in failure analysis**: The paper mentions that it "conducts a line-by-line analysis of failed model-generated submissions." Could you clarify whether this data is available for inspection, and if so, how it can be accessed?

Regarding **Finding 3**, I am somewhat confused. The result seems to align with standard expectations from pass\@k estimation, as seen in the [code\_eval implementation](https://github.com/huggingface/evaluate/blob/main/metrics/code_eval/code_eval.py#L198): specifically, increasing the number of samples tends to improve scores, while increasing problem difficulty results in smaller gains. Could you elaborate on what the novel insight is here?

The authors argue that Codeforces tags are noisy due to their crowd-sourced nature and therefore re-annotate all problems manually using expert programmers. However, are the manually annotated tags significantly different from the original Codeforces tags? It would strengthen the paper to include a quantitative analysis demonstrating how this re-annotation impacts the results, to justify the necessity of the manual effort.

**Strengths Contributions:**

1. The authors' efforts to mitigate data leakage through live contest evaluation.
2. The paper clearly distinguishes its live and challenging benchmark.
3. The paper is organized, and easy to understand.
4. The tag-wise analysis is a useful direction for evaluating LLM performance in a more granular way.

---

> ### Author Rebuttal · Authors · 2025-07-30
>
> #### Q1: Lack of metadata: There appears to be no information about problem categories or tags.
>
> Thank you for pointing this out. We have included metadata information for our benchmark in Table 2, which provides details on problem counts, their category distributions, and average Codeforces difficulty ratings. These statistics aim to give a clear overview of the problem diversity and difficulty across the benchmark.
>
> ---
>
> #### Q2: Limited source diversity: The dataset only includes problems from Codeforces and QOJ.
>
> Thank you for the insightful suggestion, but here we respectfully disagree with this point - We believe LiveCodeBench Pro already delivers complete coverage of premier competitive programming tasks. We choose Codeforces for its unrivaled reputation in rigorous problem setting—nearly every active contest setter today publishes on Codeforces, ensuring both depth and task quality. QOJ complements this by pulling from a remarkably diverse mix of sources, so together they cover the full spectrum of premier competitive‑programming tasks without diluting our benchmark with lesser‑vetted problems. Including smaller judges or less well-known problem sources without rigorous test‑validation or editorial review would only introduce poorly specified tasks and dilute our benchmark’s fidelity—so we focus on sources with guaranteed quality, and will revisit others as they mature.
>
> ---
>
> #### Q3: Missing ELO scores: No difficulty rating (e.g., ELO score) is available for the problems.
>
> We have provided per-problem ELO difficulty scores as part of our benchmark, which serve as the basis for our classification into Easy, Medium, and Hard categories. These difficulty ratings are also summarized by tag in Table 2, offering a comprehensive view of problem difficulty across different categories.
>
> ---
>
> #### Q4: Absence of LLM-generated answers: It is unclear whether large language model (LLM) outputs are included.
>
> We have included one LLM-generated output, specifically from DeepSeek R1 and GPT-o4-mini-high in the Appendix Section L Case Studies to showcase their reasoning processes. As for code outputs, we plan to release the complete set to ensure transparency and support further analysis by the community.
>
> ---
>
> #### Q5: No token-level generation data: There is no record of token-by-token generation traces from the LLMs.
>
> Appendix Section L Case Studies to showcase their reasoning processes. We also annotate with human expert to explain each reasoning process steps.
>
> ---
>
> #### Q6: Missing inference cost: The cost (e.g., token count, latency) per LLM answer is not reported.
>
> We report the approximate dollar cost per problem in the last column of Table 1, providing a standardized estimation based on the pricing of the corresponding API provider. This offers a fair comparison of inference cost across different models.
>
> ---
>
> #### Q7: Ambiguity in failure analysis: The paper mentions that it "conducts a line-by-line analysis of failed model-generated submissions." Could you clarify whether this data is available for inspection, and if so, how it can be accessed?
>
> Thank you for your concern. We have compiled the solutions and error analyses for all the evaluated problems, and we will release them along with the code and dataset. We hope this will be helpful and inspiring to the community.
>
> ---
>
> #### Q8: Regarding Finding 3, I am somewhat confused. The result seems to align with standard expectations from pass@k estimation, as seen in the code_eval implementation: specifically, increasing the number of samples tends to improve scores, while increasing problem difficulty results in smaller gains. Could you elaborate on what the novel insight is here?
>
> This is an excellent point. We agree that the pass@k strategy can effectively assist LLMs on problems within or near their capability range, such as those in our Medium and Easy splits. However, for problems that lie beyond the model’s current abilities (e.g., our Hard split), we observe that increasing k does not yield any improvement. In such cases, pass@k offers little to no benefit, highlighting the true limitations of current models.
>
> ---
>
> #### Q9: The authors argue that Codeforces tags are noisy due to their crowd-sourced nature and therefore re-annotate all problems manually using expert programmers. However, are the manually annotated tags significantly different from the original Codeforces tags? It would strengthen the paper to include a quantitative analysis demonstrating how this re-annotation impacts the results, to justify the necessity of the manual effort.
>
> Thank you for this excellent question regarding our re-annotation efforts. While we cited the crowd-sourced nature of Codeforces tags as a primary motivator (L558-561), our decision was also data-driven. A key issue is tag sparsity: 17 of the 35 original tags appeared in less than 4% of problems, making robust statistical analysis difficult. Our manual annotation ensures all categories have sufficient prevalence. Furthermore, an internal analysis revealed the original tags' lower quality. On average, we found 1.2 discrepancies per problem between our expert labels and Codeforces tags, compared to just 0.6 discrepancies among our expert annotators. This is often because Codeforces tags can be overly inclusive. Crucially, the original set omitted key tags like "ad-hoc," "case work," and "segment tree," which are vital for analyzing LLM performance on specific reasoning patterns.

---

> > ### Comment · Reviewer_NmLn · 2025-08-02
> >
> > Follow Q1:
> > Thank you for the clarification. I understand that the metadata is provided in Table 2. However, I couldn't find categories, tags, or cognitive focus in the Hugging Face dataset provided. Could you confirm if they are included or if they are available elsewhere?
> >
> > Follow Q2:
> > I'm a bit confused regarding the sources. The paper mentions ICPC/IOI/THUPC, but the source listed is QOJ. Did you use the QOJ subset for these contests, or did you collect directly from ICPC/IOI/THUPC?
> >
> > Follow Q3:
> > I noticed the ELO scores in Table 2, but I couldn't find them in the provided dataset. Could you clarify if they are part of the release?
> >
> > Follow Q8:
> > If I understand correctly, the key insight is that the hard split is really hard, and the current model struggles with it, even as k increases. Do I have that right?
> >
> > Follow Q9:
> > Could you clarify how the expert labels were compared to the original labels? Since different classification methods were used, how did you measure the discrepancies between them?

---

> > > ### Author Response · Authors · 2025-08-02
> > >
> > > We appreciate your response and your attention to our rebuttal!
> > >
> > > #### Follow Q1: Thank you for the clarification. I understand that the metadata is provided in Table 2. However, I couldn't find categories, tags, or cognitive focus in the Hugging Face dataset provided. Could you confirm if they are included or if they are available elsewhere?
> > >
> > > Thank you for your question. Our dataset is continuously updated with new live problems being added on a rolling basis. Given the dynamic nature of the dataset and the large volume of incoming problems, it is not feasible for us to manually annotate each new entry with categories, tags, or cognitive focus. To ensure consistency and maintainability, we have chosen not to include these annotations in the main Hugging Face dataset release. However, we have created a separate, static subset of the dataset in which categories, tags, and cognitive focus have been carefully annotated. This annotated subset will be made available through a separate link, which we will share upon publication in the future for sure.
> > >
> > > ---
> > >
> > > #### Follow Q2: I'm a bit confused regarding the sources. The paper mentions ICPC/IOI/THUPC, but the source listed is QOJ. Did you use the QOJ subset for these contests, or did you collect directly from ICPC/IOI/THUPC?
> > >
> > > Thank you for the question. Most premium contests such as ICPC, IOI, and THUPC typically release their official problem statements and test datasets, but do not provide a public online judge to reproduce results directly. QOJ serves as a centralized platform that hosts a large portion of ICPC/IOI/THUPC contests using the official problem statements and test data. We cited QOJ as the source because it offers an easily reproducible setup for users, without the need to manually configure a full-fledged online judging environment, which can be non-trivial to set up correctly.
> > >
> > > ---
> > >
> > > #### Follow Q3: I noticed the ELO scores in Table 2, but I couldn't find them in the provided dataset. Could you clarify if they are part of the release?
> > >
> > > Thank you very much for pointing this out. We will update this part in a future revision. For now, our dataset includes a difficulty split, which already separates problems of varying difficulty levels.
> > >
> > > ---
> > >
> > > #### Follow Q8: If I understand correctly, the key insight is that the hard split is really hard, and the current model struggles with it, even as k increases. Do I have that right?
> > >
> > > Yes, that’s absolutely correct! Your understanding is spot on. For the medium and easy splits, we observed that the pass@k strategy significantly improves model performance—as shown in Figure 4, the pass@k score for o4-mini-low increases from 18% to 40%. However, for the hard split, the model still fails to solve a single problem, even as k increases. We believe this indicates that the hard split is well beyond the model’s current capability, to the point where it cannot even guess the correct answer by chance.
> > >
> > > ---
> > >
> > > #### Follow Q9: Could you clarify how the expert labels were compared to the original labels? Since different classification methods were used, how did you measure the discrepancies between them?
> > >
> > > Thank you for this excellent follow-up question regarding our comparison methodology. Our analysis focused exclusively on the 16 tags that overlap between our refined 19-tag schema and the original 35 Codeforces tags. For any given problem, we defined a "discrepancy" as the count of these shared tags that was applied by one party but not the other, leading to our reported average of 1.2 discrepancies per problem. We also wish to clarify that our tag set was developed independently, not by simply curating from Codeforces. We began with an initial set of 29 tags, which we consolidated by merging related concepts and removing categories that were ambiguously defined (e.g., "Set/Map") or too infrequent for robust analysis (e.g., "Hash"). This ensured our final labels were both meaningful and statistically sound.

---

> > > > ### Comment · Reviewer_NmLn · 2025-08-04
> > > >
> > > > Follow Q2: Thank you for the clarification. However, I think my original question wasn't fully addressed. If QOJ already hosts these contests, did you simply use the problems and test data directly from QOJ, or did you also collect data independently from ICPC/IOI/THUPC sources?
> > > >
> > > > If QOJ’s coverage is incomplete, did you supplement it with additional data?
> > > > If QOJ’s coverage is complete, what was the motivation for collecting the data again?
> > > >
> > > > More broadly, what specific efforts or contributions did your make regarding dataset curation, given that QOJ already provides a fairly complete setup?

---

> > > > > ### Author Response · Authors · 2025-08-04
> > > > >
> > > > > #### Follow Q2-1: Thank you for the clarification. However, I think my original question wasn't fully addressed. If QOJ already hosts these contests, did you simply use the problems and test data directly from QOJ, or did you also collect data independently from ICPC/IOI/THUPC sources?
> > > > >
> > > > > Thanks for your follow-up question! We do not directly copy from QOJ. While QOJ covers many contest problems, we don’t just reuse its content. We also collect problems and test data independently from official sources such as the ICPC, IOI series, and THUPC websites, online judges. We prefer original, high-quality sources, especially when QOJ versions are incomplete, poorly formatted, or lack critical metadata.
> > > > >
> > > > > ---
> > > > >
> > > > > #### Follow Q2-2: If QOJ’s coverage is incomplete, did you supplement it with additional data? If QOJ’s coverage is complete, what was the motivation for collecting the data again?
> > > > >
> > > > > Thanks for asking about QOJ's coverage limitations. While QOJ's coverage appears extensive, it suffers from significant gaps and quality issues that necessitate independent data collection. Even when coverage appears sufficient, the data quality is inconsistent. Many problems appear in hard-to-parse formats: embedded LaTeX that renders incorrectly, scanned PDFs with poor OCR accuracy, or image-based mathematical expressions. Critical metadata is systematically absent, including contest dates and official difficulty classifications. These limitations, combined with QOJ's lack of standardized problem categorization, make it unsuitable as a foundation for rigorous LLM evaluation research.
> > > > >
> > > > > ---
> > > > >
> > > > >
> > > > > #### Follow Q2-3: More broadly, what specific efforts or contributions did your make regarding dataset curation, given that QOJ already provides a fairly complete setup?
> > > > >
> > > > > Thank you for this excellent question about our unique contributions. Our contributions extend far beyond data aggregation to create a research-ready benchmark. We carefully curate problems from premier 2025 contests (ICPC, IOI series, THUPC), standardize all formatting into machine-readable structures, and reconstruct complete test suites with proper graders when necessary. Most importantly, we introduce comprehensive expert annotation through our novel Cognitive and Algorithmic Tag system. Each problem receives manual tagging from knowledge-heavy to observation heavy. This expert-curated metadata, entirely absent from existing sources, enables fine-grained analysis of LLM problem-solving capabilities and represents the key innovation that transforms raw contest problems into a valuable research benchmark.

---

> > > > > > ### Comment · Reviewer_NmLn · 2025-08-06
> > > > > >
> > > > > > Could the authors provide arguments to support your efforts independently of official sources, perhaps through quantitative experiments or specific examples (due to time constraints)? You clarify that 'Even when coverage appears sufficient, the data quality is inconsistent. Many issues arise from hard-to-parse formats, such as embedded LaTeX that renders incorrectly, scanned PDFs with poor OCR accuracy, or image-based mathematical expressions' in QOJ. Do these problems also exist in official sources? How much of the gap in data quality have you addressed?

---

> > > > > > > ### Author Response · Authors · 2025-08-06
> > > > > > >
> > > > > > > Thank you for the excellent question. We address a key trade-off in benchmark creation: while official sources offer high-quality problems, they are scattered and lack standardization; aggregator platforms like QOJ centralize content but often compromise on data quality. Our work bridges this gap.
> > > > > > >
> > > > > > > Over 70% of QOJ problems required manual PDF-to-digital conversion. More importantly, we invested more in expert-driven curation for selecting recent problems, annotating them with Olympiad-level insights, and building a robust evaluation pipeline.
> > > > > > >
> > > > > > > This combination of rigorous standardization and expert annotation is absent from existing sources and forms the foundation for our novel findings. Importantly, our contribution goes beyond dataset creation to include deep expert analysis of model behavior.

---

> > > > > > > > ### Comment · Reviewer_NmLn · 2025-08-07
> > > > > > > >
> > > > > > > > The author's response solved many of my confusions, and I improved my score.

---

### Official Review · Reviewer_9JBL · 2025-06-23

**Rating:** 5
**Confidence:** 4

**Summary:**

This paper presents LiveCodeBench Pro which awkwardly reveals that current LLMs struggle in problems that require complex algorithmic reasoning. Those problems, however, could be easily solved by elite humans. Through a comprehensive evaluation of different LLMs, authors present a few conclusions and findings that might help the community to build more powerful code-centric LLMs.

**Dataset Code Accessibility:**

Partly

**Dataset Code Comments:**

The code is in good shape but the evaluation implementation on LLM answers is missing.

**Ethical Considerations:**

No, there are no or only very minor ethics concerns

**Final Justification:**

Thank you for the great work and explanation, and I appreciate the efforts of making the platform open. I decided to maintain the rating.

**Limitations Weaknesses:**

1. Authors might want to assume the readers / reviewers with zero background knowledge of competitive programming tests. However, readers without such expertise will find it difficult to understand the core setting of this work, e.g. the taxonomy design. See my next question.

2. The boundary of observation-heavy problems and knowledge-heavy problems seems blurry for me – Say two examples proposed in L114 and L147. Two problems seem to both need the model to figure out the connection between concepts in the problem and common programming techniques. Then how should the human annotators tell the difference between these two categories?
    - Note I won’t consider this as a weakness in this work, but rather proposing a question. This did not affect my general evaluation of this work.

3. The attached code repository seems to only have the interfaces to submit the questions to LLMs. Do the authors have the plan to release the evaluation code/platform as well? I assume it will resemble the online judge (OJ) system used in programming contests.

**Strengths Contributions:**

1. Interesting (but expected) findings in the work:
  - Reasoning models are generally better than non-reasoning ones at the cost of higher spending per problem.
  - LLMs are good at knowledge-intensive problems – thanks to the parametric memory.
  - Pass@k generally reflects the reasoning boundaries of LLMs, measured by the increased but saturated performance as k grows.
  - Reasoning boosts the performance in knowledge-intensive problems, possibly because the structured thinking in the reasoning process – I expect the model will restate the core techniques and templates to use that would help solve the problem.

2. The benchmark designed moderated by human competitors ensures its quality.
3. Authors developed a taxonomy of defining competitive programming problems into knowledge-intensive, logic-heavy and observation-heavy problems.

---

> ### Author Rebuttal · Authors · 2025-07-30
>
> #### Q1&Q2: Authors might want to assume the readers / reviewers with zero background knowledge of competitive programming tests. However, readers without such expertise will find it difficult to understand the core setting of this work, e.g. the taxonomy design. See my next question. The boundary of observation-heavy problems and knowledge-heavy problems seems blurry for me – Say two examples proposed in L114 and L147. Two problems seem to both need the model to figure out the connection between concepts in the problem and common programming techniques. Then how should the human annotators tell the difference between these two categories?
>
> We appreciate you raising this nuanced point. The distinction between knowledge-heavy and observation-heavy problems lies in whether the solution hinges on recalling established techniques versus discovering novel, problem-specific properties. For instance, a knowledge-heavy problem like L114 requires applying a known, advanced technique (e.g., FFT for string matching), which is standard for top competitors but difficult to derive from scratch. In contrast, an observation-heavy problem like L147 requires contestants to identify a hidden pattern (e.g., a long cycle in a sequence) through hypothesis and verification, a property not found in standard algorithm handbooks. While both demand insight, knowledge-heavy problems test the application of a known toolkit, whereas observation-heavy problems test the ability to uncover emergent patterns through ad-hoc reasoning, much like solving a puzzle.
>
> ---
>
> #### Q3: The attached code repository seems to only have the interfaces to submit the questions to LLMs. Do the authors have the plan to release the evaluation code/platform as well? I assume it will resemble the online judge (OJ) system used in programming contests.
>
> Yes, we plan to release all evaluation code and components to ensure transparency and full reproducibility. The benchmark will be continuously updated with real-time evaluation results, which is precisely what the “live” in LiveCodeBench Pro stands for. We appreciate your attention to this aspect and are committed to maintaining an open and evolving evaluation platform.

---

> > ### Comment · Reviewer_9JBL · 2025-08-01
> >
> > Thank you for the great work and explanation, and I appreciate the efforts of making the platform open. I decided to maintain the rating.

---

> > > ### Author Response · Authors · 2025-08-02
> > >
> > > Thank you very much for recognizing our work and for your constructive feedback. it has been extremely helpful to us!

---

### Official Review · Reviewer_UmKh · 2025-06-24

**Rating:** 5
**Confidence:** 3

**Summary:**

This paper introduces ​LiveCodeBench Pro, a challenging LLM programming benchmark consisting of international algorithmic competition contests. The authors' primary objective is to ​re-examine the distinctions between human and LLM capabilities in handling such algorithmic tasks.​​ To achieve this, they meticulously annotate each problem based on its algorithmic category and perform a detailed line-by-line analysis of failed model-generated code submissions. Furthermore, the authors highlight interesting phenomena regarding ​how the category and difficulty level of the tasks influence LLMs' coding performance.​​

**Dataset Code Accessibility:**

Yes

**Ethical Considerations:**

No, there are no or only very minor ethics concerns

**Final Justification:**

The rebuttal have well addressed my concerns and I recommend for acceptance.

**Limitations Weaknesses:**

1. ​Potential Limited Generalizability:​​ While the focus on high-difficulty competition problems is a ​key strength​ for probing the limits of LLMs, it might also represent a ​limitation​ regarding broader applicability. The benchmark primarily targets problems where expert humans excel. It would be valuable to discuss how these findings translate to more common, everyday programming tasks encountered by developers.

2. ​Sustainability of Expert Annotation:​​ The detailed annotation and analysis by Olympiad medalists is ​impressive and insightful. However, this process is ​highly resource-intensive​ and require human experts. If the benchmark bank gets bigger in the future, will it still be possible to annotate them one by one in this way? Is there a way to partially automate this process?

**Strengths Contributions:**

1. The paper makes a key contribution by introducing ​LiveCodeBench Pro, a significant update to existing programming benchmarks. It features much harder problems compared to existing ones.

2.  Expert Problem Annotation:​​ Another highlight is that ​LiveCodeBench Pro is annotated by ​Olympiad medalists, making this benchmark highly convincing and well-justified.

3. ​The developed tagging system (knowledge/observation/logic-heavy) is ​highly relevant and well-justified. It effectively pinpoints areas where LLMs struggle the most.

4. The paper uncovers important new findings on the new benchmark. LLMs show ​strength on implementation-heavy tasks​ but exhibit ​critical weaknesses in nuanced algorithmic reasoning and complex case analysis, especially within the identified observation-heavy problems. This opens some new revenue for the comnunity.

---

> ### Author Rebuttal · Authors · 2025-07-30
>
> #### Q1: Potential Limited Generalizability:​​ While the focus on high-difficulty competition problems is a ​key strength​ for probing the limits of LLMs, it might also represent a ​limitation​ regarding broader applicability. The benchmark primarily targets problems where expert humans excel. It would be valuable to discuss how these findings translate to more common, everyday programming tasks encountered by developers.
>
> LiveCodeBench Pro is intentionally designed to probe the upper limits of LLM capabilities by focusing on highly challenging, competition-level problems. While we agree that generality and real-world applicability are important, it's worth noting that benchmarks such as SWE-bench already aim to evaluate how well AI systems can resolve practical GitHub issues, yet LLMs still struggle on these fronts. Our goal is to contribute complementary insights by pushing the boundaries of reasoning and problem-solving, offering a different but equally valuable perspective for the community.
>
> ---
>
> #### Q2: Sustainability of Expert Annotation:​​ The detailed annotation and analysis by Olympiad medalists is ​impressive and insightful. However, this process is ​highly resource-intensive​ and require human experts. If the benchmark bank gets bigger in the future, will it still be possible to annotate them one by one in this way? Is there a way to partially automate this process?
>
> Thank you for raising this important point. We acknowledge that expert annotations, especially those provided by Olympiad medalists, are resource-intensive and may not scale easily. However, we do not believe this presents a fatal limitation. 1) The most time-consuming aspect of annotation lies in analyzing the reasons behind model errors. For this component, introducing partial automation—such as clustering error types or using LLM-assisted tagging—could be a promising direction. Furthermore, since current LLM performance on these problems has not shown rapid progress, we expect the insights derived from expert analyses to remain relevant and stable for some time. 2) For other aspects like test case generation and scoring, we agree that automation will be crucial as the benchmark grows. We are actively exploring ways to streamline these processes in future work. We appreciate your thoughtful suggestion.

---

> > ### Comment · Reviewer_UmKh · 2025-08-05
> > **Official Comment**
> >
> > The author's rebuttal has effectively addressed my concern. Therefore, I'm happy to maintain my score.

---

> > > ### Author Response · Authors · 2025-08-05
> > >
> > > We really thank the reviewer for the valuable suggestions on generalizability and annotation sustainability, which helps us improve the paper for the future version.

---

### Official Review · Reviewer_dZrw · 2025-07-03

**Rating:** 4
**Confidence:** 4

**Summary:**

The paper presents an updated version of LiveCodeBench, a benchmark consisting of real-world, challenging programming and algorithmic thinking problems drawn from CodeForces, ICPC, and IOI. A key strength of this benchmark is that it is continuously updated in near real-time, which helps mitigate data contamination and offers a cleaner, more realistic setting for evaluating the performance of large language models (LLMs). Evaluation results show that even the strongest reasoning-focused model (GPT-4.1) achieves only 53% pass@1 accuracy on medium-level problems and 0% on hard problems. Non-reasoning models perform significantly worse, often achieving single-digit pass@1 accuracy on medium problems and also failing completely on hard tasks. The benchmark includes 584 problems assembled before any accepted solutions exist publicly. Evaluation also considered the cognitive focus of each problem (logic, observation, knowledge). Results reveal that LLMs perform better on logic-heavy and knowledge-heavy problems than on observation or creativity-driven tasks. While pass@k metrics improve model performance somewhat, especially on medium-level problems, LLMs still fail consistently on the hardest problems. Reasoning models outperform non-reasoning models overall, yet both struggle similarly with observation-based tasks.

**Additional Feedback:**

**Clarification Questions:**
- Could the authors clarify what is meant by the “interactive problems” domain and elaborate on the reward hacking issue noted for o3-mini-high? This seems like an interesting sub-benchmark that could benefit from more explanation in the main text rather than only in Appendix H.
- This might extend beyond the current scope, but what do the authors see as the potential impact of an expert–LLM collaborative problem-solving approach on these hard tasks? Given that LLMs appear strong at implementation but weaker at algorithmic planning, could a collaborative process significantly improve success rates? And if so, what specific parts of the prolem-solving process can be given to the expert and which parts to the LLM, to ensure some success on the hard category of problems?

**Dataset Code Accessibility:**

Yes

**Dataset Code Comments:**

The dataset is accessible via hugging face and the README provides comprehensive guide to extract the data and analyze/evaluate it via a custom LLM interface.

**Ethical Considerations:**

No, there are no or only very minor ethics concerns

**Final Justification:**

The authors have sufficiently addressed my queries. I have increased my score to borderline accept with a higher confidence rating.I think the dataset addresses a timely concern about real-world evaluation of LLMs and their results highlight that while the LLMs are quite good at implementation, there needs to be further work on improving their planning capabilities. The results in this work also align with recent similar findings in other domains (such as CHI). I also liked the inclusion of different problem types (such as interactive problems) which go a bit beyond the traditional datasets which use typical program synthesis/debugging problems.

**Limitations Weaknesses:**

**Weaknesses:**
- While the paper shows that LLMs fail on hard and observation-heavy tasks, it would be valuable to include a deeper error analysis to understand “why”. For instance, do the models fail at the planning stage, make early logical mistakes, or fail to recognize critical observations?
- Exploring whether adjusting sampling temperature or other decoding parameters helps the models explore alternative solution trajectories at least to begin with—especially on the hard problems—could add meaningful depth.
- Beyond just reporting a 0% success rate on the hard tier, it would strengthen the contribution to analyze whether LLMs at least partially approach the correct logic or if they are completely off base.
- A brief discussion on the potential cognitive limitations of current LLM architectures—particularly around structured reasoning and heuristic-driven problem solving—would help contextualize the findings.

**Strengths Contributions:**

**Strengths:**
- The continuous refreshing of the benchmark significantly reduces the risk of training data contamination, making it a practical and trustworthy evaluation tool.
- The paper shows that even advanced LLMs remain far from AGI, especially on creative or observation-heavy tasks, reinforcing the role of human problem-solving skills.
- The evaluation is comprehensive and the paper is clearly written.
- The discussion around pass@k metrics is insightful, illustrating that increased sampling helps slightly on moderately difficult tasks but does not close the gap on truly hard problems.
- Using ELO ratings to contextualize LLM performance relative to human participants is a useful addition, making the results more interpretable.

---

> ### Author Rebuttal · Authors · 2025-07-30
>
> #### Q1: While the paper shows that LLMs fail on hard and observation-heavy tasks, it would be valuable to include a deeper error analysis to understand “why”. For instance, do the models fail at the planning stage, make early logical mistakes, or fail to recognize critical observations?
>
> Thank you for this insightful suggestion regarding a deeper error analysis. Our findings indicate that the vast majority of failures on hard and observation-heavy tasks, approximately 80-90%, stem from high-level reasoning deficits rather than implementation errors. These include fundamental flaws in algorithmic planning, early logical missteps, or a failure to identify critical, problem-specific observations. The primary failure mode often correlates with the problem category: for observation-heavy tasks, models typically miss the key insight, whereas for other hard problems, the breakdown occurs during the strategic planning phase. While a more granular breakdown is valuable, providing precise, generalizable statistics is challenging as the root cause is highly dependent on problem type and difficulty. Given the labor-intensive nature of expert annotation, a large-scale quantitative study of specific error sub-types remains an important direction for future work.
>
> ---
>
> #### Q2: Exploring whether adjusting sampling temperature or other decoding parameters helps the models explore alternative solution trajectories at least to begin with—especially on the hard problems—could add meaningful depth.
>
> We test the performance of GPT-4.1 under different temperature settings across three sets. The results are shown in the table below. We did not observe a significant correlation between the sampling temperature and performance. We will include this discussion in the future version.
>
> |GPT-4.1 Temp |Set 1 |Set 2|Set 3|
> |:-:|:-:|:-:|:-:|
> |0.2| 19.7% | 26.4% | 15.7% |
> |0.5| 15.5% | 25.5% | 19.1% |
> |1.0| 22.5% | 30.0% | 21.3% |
>
> ---
>
> #### Q3: Beyond just reporting a 0% success rate on the hard tier, it would strengthen the contribution to analyze whether LLMs at least partially approach the correct logic or if they are completely off base.
>
> This is an excellent question regarding the proximity of LLM attempts to a correct solution. Based on our expert analysis of the hard-tier problems, we found that in over 90% of failure cases, the models diverge from the correct path during the initial observation or planning phase. This typically manifests as pursuing a flawed logical premise or implementing an algorithm with the wrong time complexity. In a small minority of cases, particularly on topics like combinatorics or dynamic programming, a model might guess a correct high-level approach, likely through pattern matching from its training data rather than chain-of-thought reasoning. However, even when the initial idea is directionally correct, these attempts invariably fail during implementation due to the intricate details and edge cases, ultimately leading to no successful submissions.
>
> ---
>
> #### Q4: A brief discussion on the potential cognitive limitations of current LLM architectures—particularly around structured reasoning and heuristic-driven problem solving—would help contextualize the findings.
>
> Thank you for the suggestion. Specifically, LiveCodeBench Pro reveals that while LLMs perform reasonably well on knowledge-intensive problems, they consistently struggle with observation-heavy tasks that require structured reasoning and heuristic-driven problem solving. Our analysis shows that failures often stem from idea-level flaws rather than implementation bugs, highlighting limitations in the model’s ability to generate creative insights and handle complex edge cases. We believe this underscores a fundamental gap in current architectures and motivates future research toward more reasoning-capable systems.
>
> ---
>
> #### Q5: Could the authors clarify what is meant by the “interactive problems” domain and elaborate on the reward hacking issue noted for o3-mini-high? This seems like an interesting sub-benchmark that could benefit from more explanation in the main text rather than only in Appendix H.
>
> Thank you for the opportunity to clarify the nature of interactive problems and the observed reward hacking. Unlike traditional problems with a single input-output cycle, interactive problems involve a multi-turn dialogue between the solution and a judge system. A classic example is a number-guessing game where the program must adapt its strategy based on "higher" or "lower" feedback. The reward hacking we noted in o3-mini-high refers to its tendency to exploit simple heuristics, such as echoing the judge's input directly as output. We hypothesize this stems from challenges in assigning accurate credit during its RL phase for this format, as this simple heuristic can sometimes pass basic test cases. It is noteworthy that this behavior is absent in newer OpenAI models, suggesting the underlying reward model has been refined.
>
> ---
>
> #### Q6: This might extend beyond the current scope, but what do the authors see as the potential impact of an expert–LLM collaborative problem-solving approach on these hard tasks? Given that LLMs appear strong at implementation but weaker at algorithmic planning, could a collaborative process significantly improve success rates? And if so, what specific parts of the prolem-solving process can be given to the expert and which parts to the LLM, to ensure some success on the hard category of problems?
>
> Thank you for this insightful question. We agree that an expert-LLM collaborative framework holds immense potential for solving hard problems. We envision a synergistic division of labor where the human expert leverages their intuition and strategic thinking to guide the overall approach. This includes identifying critical observations, specifying the algorithmic paradigm (e.g., dynamic programming, greedy), and defining complex case analyses. The LLM would then excel at translating these high-level plans into accurate and efficient code. It could also assist the expert by generating extensive test cases to help uncover patterns or by automatically debugging the implementation against a brute-force solution. This collaborative process, which leverages the complementary strengths of both human experts and LLMs, could substantially enhance problem-solving performance, potentially leading to a significant improvement on these challenging problems.

---

> > ### Comment · Reviewer_dZrw · 2025-08-06
> > **Thank you for the Rebuttal!**
> >
> > Thank you authors, for the comprehensive response to all the queries.
> > - Thanks for the additonal results using GPT models with different temperature parameters. It seems like simply increasing temperture does not help explore diverse solution pathways. As I understand now, the complexity stems from exploring diverse "planning" pathways which cannot be controlled by only altering the temperature in the current setup.
> > - It would be good to include a short discussion inthe final paper around this distinction between the planning and implementation stage requirements for the tasks -- and that LLMs are primarily struggling in the planning stage which causes a drop in performance. This seems quite central to the poor performance on the hard tasks.
> > - Here are two relevant recent paper that the authors might find useful, related to this discussion:
> > Shi et al. When Models Know More Than They Can Explain: Quantifying Knowledge Transfer in Human-AI Collaboration,
> > Si et al. The Ideation–Execution Gap: Execution Outcomes of LLM-Generated versus Human Research Idea

---

> > ### Author Response · Authors · 2025-08-06
> >
> > Thanks for your follow-up questions!
> >
> > #### Q1: Thanks for the additonal results using GPT models with different temperature parameters. It seems like simply increasing temperture does not help explore diverse solution pathways. As I understand now, the complexity stems from exploring diverse "planning" pathways which cannot be controlled by only altering the temperature in the current setup.
> >
> > We sincerely appreciate your insightful comment. We fully agree with your observation merely increasing the temperature does not sufficiently promote exploration of diverse solution pathways. In our experiments, we also found that reasoning models tend to perform best under their default temperature settings; deviating from these often leads to noticeable performance drops. This suggests that the relation between temperature and diversity in reasoning tasks is more nuanced than expected, and we believe this warrants further investigation in future work.
> >
> > ---
> >
> > #### Q2: It would be good to include a short discussion inthe final paper around this distinction between the planning and implementation stage requirements for the tasks -- and that LLMs are primarily struggling in the planning stage which causes a drop in performance. This seems quite central to the poor performance on the hard tasks.
> >
> > Thank you for highlighting this important distinction. We agree that the performance bottleneck for large language models in our setting primarily lies in the planning stage rather than the execution stage. This insight aligns with our own observations: while models are often capable of implementing a given plan reliably, they struggle to generate effective multi-step plans in the first place. We will add a brief discussion of this distinction in the revised paper, emphasizing that the planning-execution gap plays a central role in the difficulty of the harder tasks.
> >
> > ---
> >
> > #### Q3: Here are two relevant recent paper that the authors might find useful, related to this discussion: Shi et al. When Models Know More Than They Can Explain: Quantifying Knowledge Transfer in Human-AI Collaboration, Si et al. The Ideation–Execution Gap: Execution Outcomes of LLM-Generated versus Human Research Idea
> >
> > Thank you very much for the helpful references. We found both highly relevant and insightful. The notion of a gap between ideation/planning and execution resonates well with our findings, and we believe this line of work further supports our interpretation of the performance bottleneck. We will cite both works in the updated version of the paper and briefly discussed how they relate to our setting.

---

### Note · Authors · 2025-08-13

We thank the AC and reviewers for thoughtful, constructive feedback. We are encouraged by the recognition of: (i) a live, contamination-resistant benchmark; (ii) Olympiad-annotated cognitive taxonomy; (iii) clear evaluation with ELO context; and (iv) the key result that frontier LLMs fail on the hardest problems primarily due to planning not implementation.

We summarize the rebuttal process and glad to see the following concerns have been addressed:

- Provided deeper failure analysis: most hard/observation-heavy errors are idea-/planning-stage (≈80–90%); we will foreground the planning–execution gap.

- Ran temperature ablations: no material gains; reasoning models work best near defaults.

- Clarified pass@k: helpful on easy/medium, saturates on hard, which leads to genuine capability limits.

- Explained interactive problems and expand the appendix.

- Detailed curation: beyond QOJ, we integrate official ICPC/IOI/THUPC sources, standardize formats/graders, and add expert metadata.

- Committed releases: a static annotated subset (categories/tags/cognitive focus), per-problem difficulty/ELO, evaluation code, failure analyses, and LLM outputs.

We sincerely thank reviewers again for their meaningful discussion!

---

### Decision · Program_Chairs · 2025-09-18

**Decision:**

Accept (poster)

**Comment:**

Strengths & Contributions
- Introduces a continuously updated, real-world programming benchmark (from CodeForces, ICPC, IOI) that mitigates data contamination and raises the difficulty level compared to existing benchmarks.
- Features expert annotation by Olympiad medalists and a taxonomy (knowledge / logic / observation-heavy), providing a rigorous framework for analysis.
- Provides a comprehensive evaluation across reasoning vs. non-reasoning LLMs, pass@k metrics, and ELO ratings relative to human competitors.
- Reveals key findings:
Reasoning-focused models (e.g., GPT-4.1) significantly outperform non-reasoning ones but still fail on hard/observation-heavy tasks (53% on medium, 0% on hard).
LLMs are stronger on knowledge-heavy and implementation-heavy tasks, but weak on nuanced algorithmic reasoning and observation-driven problem solving.
Pass@k helps slightly on medium problems but does not close the gap on harder tasks.

Weaknesses / Limitations
- Lack of deeper error analysis: unclear if failures stem from flawed planning, logic breakdowns, or missed observations.
- Generalizability concerns: benchmark focuses on elite-competition problems; unclear how results translate to everyday programming tasks.
- Annotation scalability: reliance on Olympiad experts makes the process resource-intensive, raising sustainability concerns.
- Taxonomy clarity: some ambiguity between knowledge-heavy and observation-heavy categories.
- Code release: repository only provides interfaces; unclear if a full evaluation platform will be shared.

===== FINAL UPDATE FROM DB Track PCs ====

The final decision for this paper has been taken by the program chairs after consultation with the SACs. All Senior Area Chairs have ranked papers according to the feedback from the AC during the review process. We decided to leave the original meta-review to reflect the opinion of the AC in light of the initial discussions with reviewers and SAC.